# Multi-Evaluating Strategy for Siji-kangbingdu Mixture: Chemical Profiling, Fingerprint Characterization, and Quantitative Analysis

**DOI:** 10.3390/molecules24193545

**Published:** 2019-09-30

**Authors:** Zhuoru Yao, Jingao Yu, Zhishu Tang, Hongbo Liu, Kaihua Ruan, Zhongxing Song, Yanru Liu, Kun Yan, Yan Liu, Yuping Tang, Huqiang Ma

**Affiliations:** 1Shaanxi Collaborative Innovation Center of Chinese Medicine Resources Industrialization/State Key Laboratory of Research & Development of Characteristic Qin Medicine Resources (Cultivation)/Shaanxi Innovative Drug Research Center, Shaanxi University of Chinese Medicine, Xianyang 712000, China; yaozhuoru0708@gmail.com (Z.Y.); rkh15709100891@163.com (K.R.); szx74816@sina.com (Z.S.); 1501011@sntcm.edu.cn (Y.L.); 2College of pharmacy, Shaanxi University of Chinese Medicine, Xixian New Area, Xianyang 712046, China; 18791029030@139.com (K.Y.); ly712537@163.com (Y.L.); yupingtang9@126.com (Y.T.); 3Shaanxi Haitian pharmaceutical co., LTD, Xixian New Area, Xianyang 712046, China; mahuqiang_2016@163.com

**Keywords:** LC-MS, fingerprint, QAMS, Siji-kangbingdu mixture, quality evaluation

## Abstract

Siji-kangbingdu mixture is an anti-inflammatory, anti-bacterial, and anti-viral herbal mixture which is frequently used by doctors to treat upper respiratory infections. It’s important to establish an efficient and economical quality-control method to ensure the quality consistency and efficacy stability of Siji-kangbingdu mixture. In this study, an integrated multi-evaluation method was established, sequentially involving UPLC-TripleTOF-MS analysis, UPLC fingerprint analysis, and the quantitative analysis of multi-components using the single-marker (QAMS) method. With one chromatographic condition, a total of 71 compounds were identified by MS and MS/MS information, with a mass error of less than 5 ppm; 49 peaks detected in 254 nm were selected to establish the fingerprint similarity model, and 7 chemical compounds were simultaneously determined, namely, chlorogenic acid, liquiritin, rutin, isochlorogenic acid A, forsythin, forsythoside A, and glycyrrhizic acid, with forsythoside A as the reference standard. There was no significant difference in the content of the seven compounds between the QAMS method and the external standard method (ESM). The established multi-evaluation method will largely promote the quality control and standardization process of Siji-kangbingdu mixture. It also provides a reference workflow for the overall evaluation of TCM patent medicines, from chemical profiling to fingerprint and quantitative analysis.

## 1. Introduction

Siji-kangbingdu mixture is a nonprescription herbal drug, produced through a series of procedures which are protected by a Chinese patent [1]. Siji-kangbingdu mixture is a Chinese herbal preparation in liquid form, containing 11 herbs, namely *Houttuyniae Herba*, *Platycodonis Radix*, *Mori Folium*, *Forsythiae Fructus*, *Schizonepetae Herba*, *Menthae Haplocalycis Herba*, *Perillae Folium*, *Armeniacae Semen Amarum*, *Phragmitis Rhizoma*, *Chrysanthemi Flos*, and *Glycyrrhizae Radix et Rhizoma*. It is one of the most saleable traditional Chinese Medicines (TCMs) in the Chinese drug market, manufactured by Shaanxi Haitian pharmaceutical Co., LTD. Consumers are allowed to buy it easily from any drugstore as needed. The main therapeutic effects of Siji-kangbingdu mixture are anti-inflammation, anti-infection, and fever-relief. Clinically, it is commonly used to treat influenza and upper respiratory tract infections in patients, especially children, with symptoms of headache, fever, cough, and rhinorrhea. So, the quality standards of Siji-kangbingdu mixture must be higher than common TCM formulas, whose quality standards are limited to simple chemical characterization or quantitation [2]. The low-level quality standards of TCM formulas or patent drugs have been recognized as being responsible for the unstable efficacy of TCM preparations [3]. More and more TCM scientists and regulatory officials are appealing for higher quality standards for TCM patent medicines, including comprehensive chemical profiling, strict batch consistency, and efficacy-related, multicomponent quantitation [4,5]. In contrast to its definite curative effects, the chemical profile of Siji-kangbingdu mixture is largely unclear, due to the complexity of the herbal composition and chemical diversity in each herb, which in turn, have limited the ability to perform quality evaluations and to offer a rationale for the clinical use of this herbal medicine.

A great deal of literature has reported on the in vitro and in vivo activities of chemical compounds that are possibly contained in Siji-kangbingdu mixture, including the antioxidant and anti-inflammatory effects of chlorogenic acid [6,7] and rutin [8], the therapeutic effects of liquiritin [9] in the treatment of cancer and cancer-related complications, the anti-viral effects of forsythoside A [10], Isochlorogenic acid A [11], and glycyrrhizic acid [12], and the anti-inflammatory and antibacterial effects of forsythin [13]. These compounds may be important for the efficacy of Siji-kangbingdu mixture, and may be able to serve as the quality markers (Q-makers) of it [3].

Nowadays, the standardization of TCM patent medicines has become a trend which is increasingly emphasized by the drug administration department, the core of which is to ensure the consistency of drug quality between different batches so as to maintain drug efficacy. Under this circumstance, it is an important issue to quickly establish a method with which to evaluate the chemical basis, fingerprint similarity, and compound contents of Siji-kangbingdu mixture. Some researchers have reported a quantitative analysis method for Siji-kangbingdu mixture, determining the contents of 3 to 4 compounds but leaving the chemical profile and fingerprint unknown [14,15,16].

LC-TripleTOF-MS technology has been widely used in analyzing and identifying chemical compounds in herbal medicines, with high mass accuracy, high sensitivity, and high scanning speeds. This technology can provide large data capacity and the deep detection of chemical compounds in herbal medicines. Additionally, with a compound library, we can easily search for chemical compounds in samples with satisfactory confidence levels [17,18]. The fingerprint similarity analysis method has been accepted worldwide to reflect the internal quality of herbal medicines by showing their holistic and fuzzy characteristics, which is consistent with TCM methodology [19,20,21]. The quantitative analysis of multi-components using the single-marker (QAMS) method for herbal medicine was first put forward by Professor Zhimin Wang [22,23]; this method allows efficient quantitative evaluations to be made of herbal medicines. 

In order to establish a comprehensive, effective, and economic quality evaluation method for Siji-kangbingdu mixture, we integrated LC-TripleTOF-MS technology, fingerprint characterization technology, and the QAMS method, to form a strategy adopting the advantages of all of these technologies. This strategy not only provides overall chemical information, but also comprehensively evaluates quality consistency, as well as determining various components simultaneously, which is comprehensive, fast, reliable, and economical.

## 2. Results

### 2.1. Optimization of Chromatographic Conditions

Siji-kangbingdu mixture mainly contains high-polarity compounds, and the column-retention properties of these compounds are very similar, which makes it hard to separate them from each other. The authors attempted to separate the various components using HPLC systems but failed, due to the tedious analytical time of more than 120 min. So, in this study, we employed UPLC systems that reduce the analytical time to less than 60 min. 

Considering sample stability and separation efficiency, the Siji-kangbingdu mixture samples (provided in a 150 mL brown plastic bottle in liquid form) are diluted with methanol solutions (1:1, v/v) and filtered through 0.22 μm Millipore membranes. The concentration of dilution solutions (methanol, 50% methanol, 20% methanol) was optimized, and finally, 20% methanol was selected with the best separation degree. The mobile phase and column temperature were both optimized. Methanol, acetonitrile, and methanol-acetonitrile (1:1, *v/v*) were tested as organic phases, but only acetonitrile was able to clearly separate forsythoside A from isochlorogenic acid A. Similarly, a column temperature of 30 °C was shown to be the best. In this study, the elution progress is one of the most important conditions, because, on the one hand, the intricate peaks need elaborate elution progress to separate them, while on the other, the authors aimed to use the same elution progress in all experiments. After a large number of tests, a gradient elution procedure, as provided in Section 4.2, was determined, with the largest peak capacity, the best peak resolution, and the shortest time.

The detection parameters of the UV detector and TripleTOF mass spectrometer were carefully selected. After full-wavelength scanning, 254 nm was selected for UPLC fingerprint and quantitative analysis; it was shown to provide for the best information presentation and separation degree, as well as an appropriate peak ratio. The declustering and collision energies for primary and secondary scanning were set for the best compound adaptability, as shown in Section 4.2.

### 2.2. Chemical Identification Analysis by UPLC-TripleTOF-MS Technology

As for Siji-kangbingdu mixture, no existing chemical profile has been established, which is a barrier for understanding its chemical basis, and makes it difficult to carry out quantitative studies. An alternative is that we can search for its chemical compounds in the literature and in databases, as some researchers have done; however, but there is a large degree of uncertainty about the content of the observed compounds, because some compounds may be lost after preparation. 

So, the first step of our evaluation strategy was to draw an overall chemical profile for Siji-kangbingdu mixture using UPLC-TripleTOF-MS technology. As shown in the results (Figure 1 and Appendix A), a total of 106 compound peaks were detected in positive and negative ion mode, of which 71 were identified by matching the secondary mass fragments of standard material in the TCM MS/MS library with special algorithms [24]. This method can ensure enough confidence level for each identified compound in three ways: 1) the high resolution and high accuracy of the mass spectrometry allow researchers to screen compounds with a narrow mass error (i.e., less than 5 ppm); 2) The isotope pattern matching efficiently excludes false positive identifications from the results; and 3) By introducing home-made or commercially-available mass libraries of standard materials, fast and automatic comparison between unknown MS/MS spectra and standard spectra can be realized [25]. Among the most abundant compounds, 30 are flavonoids (glycosides), 7 are phenylpropanoids, 4 are phenolic acids, and 3 are triterpenes. These chemical types represent the principal chemical basis of Siji-kangbingdu mixture. Among these identified compounds, seven were verified again with standard materials, including chlorogenic acid, liquiritin, rutin, forsythoside A, isochlorogenic acid A, forsythin, and glycyrrhizic acid. These compounds, which are believed to have clinical significance, are abundant in Siji-kangbingdu mixture, and can comprise phenolic acid, flavonoids, phenylpropanoids, and saponins in its chemical makeup. 

### 2.3. Method Validation for Fingerprint Analysis and Quantitative Analysis

#### 2.3.1. Investigation of System Applicability and Specificity

Using the LC-20ADXR system, samples and standard solutions can be separated with good peak resolution and shape, and all the 7 compounds were separated with a degree of more than 1.5, which confirms the suitability of this system for our study. To investigate whether the peaks of the 7 determined compounds were free from interference peaks, negative samples were made, as shown in the results (Figure 2). According to the literature and the TCMSP database [26], liquiritin and glycyrrhizic acid are unique compounds in *Glycyrrhizae Radix et Rhizoma*, and forsythin and forsythoside A are unique compounds in *Forsythiae Fructus*. So, the two herbs were removed from the recipe in a negative sample preparation to check the specificity of the four compounds (Figure 2c,d). Rutin and isochlorogenic acid A exist in *Mori Folium*, *Chrysanthemi Flos*, *Houttuyniae Herba*, *Forsythiae Fructus*, and *Glycyrrhizae Radix et Rhizoma*, so when these herbs were all removed from the negative sample (Figure 2e), the peak of rutin and isochlorogenic A disappeared, and no interference was observed. As for chlorogenic acid, negative samples cannot be made, because all 11 herbs in Siji-kangbingdu mixture contain this compound.

#### 2.3.2. Linearity

The seven mixed reference solutions of different concentrations were analyzed, and the calibration curves were plotted with different concentrations (X) and corresponding peak areas (Y). The results (Table 1) showed that there is a good correlation between the concentration and peak area of each compound, and that the linear range of each compound is wide enough for use, with a 64 times ratio between the high and low concentration limit. Limit of Detections (LODs) and Limit of Quantities (LOQs) were determined according to a 3:1 and 10:1 signal-to-noise ratio, respectively, as shown in the table below.

#### 2.3.3. Precision, Stability, Repeatability, and Accuracy

The level of precision was calculated by the RSDs of peak areas from six repeated determinations of sample S1. The results (Table 2) showed that the RSDs of chlorogenic acid, liquiritin, rutin, forsythoside A, isochlorogenic acid A, forsythin, and glycyrrhizic acid were 2.8%, 3.2%, 1.2%, 0.42%, 3.2%, 0.64%, and 0.14% (*n* = 6), respectively, indicating that the precision of the analytical system was satisfactory.

The stability was calculated by analyzing sample S1 at 0, 2, 4, 8, 12, and 24 h time points at room temperature; the RSDs of peak areas for chlorogenic acid, liquiritin, rutin, forsythoside A, isochlorogenic acid A, forsythin, and glycyrrhizic acid were 2.63%, 2.64%, 1.06%, 0.32%, 1.77%, 0.98%, and 0.14% (*n* = 6), respectively, indicating that the samples were stable after 24 h.

Sample S1 was prepared six times and peak areas were determined in these parallel samples. The results showed that the RSDs for chlorogenic acid, liquiritin, rutin, forsythoside A, isochlorogenic acid A, forsythin, and glycyrrhizic acid were 1.52%, 3.94%, 0.85%, 0.61%, 3.79%, 3.45%, and 0.36% (*n* = 6), respectively, indicating that the repeatability of the operations was good.

The accuracy was determined through sample recovery experiments. A certain amount of reference substances was added to sample S1, and the recovery rate was calculated using Equation (1):Recovery rate (%) = (Found amount − Known amount) × 100%/Added amount,(1)

The recovery rate of chlorogenic acid, liquiritin, rutin, forsythoside A, isochlorogenic acid A, forsythin, and glycyrrhizic acid were 99.60%, 95.11%, 104.19%, 90.87%, 99.47%, 100.76%, and 95.00% (*n* = 6), respectively, with RSDs of 2.11%, 0.34%, 1.83%, 2.66%, 1.70%, 1.86%, and 4.24% (*n* = 6), respectively.

### 2.4. Quality Evaluation of Siji-kangbingdu mixture by UPLC Fingerprint Similarity Analysis and Clustering Analysis

The second step of our evaluation strategy was to give a global parameter to Siji-kangbingdu mixture with which to easily characterize its internal quality and the quality variants among different sample batches. UPLC fingerprint analysis is currently the best choice [27]. In this study, the LC-20ADXR system was used, given the advantages of this system [28].

Fifteen batches (S1–S15) of Siji-kangbingdu mixture were analyzed, and chromatograms under 254 nm were recorded. Then, the chromatogram data (CDF format) from the LC-20ADXR UPLC system were imported into the “Similarity Evaluation System of Traditional Chinese Medicine Chromatographic Fingerprint” software (2012.130723 edition, developed by the Chinese Pharmacopoeia Commission), recommended by the China Pharmacopoeia Committee. The chromatogram of S1 was set as the reference spectrum, and a total of 49 common peaks were aligned and marked manually, including the seven compounds identified in the UPLC-TripleTOF-MS analysis. The chromatographic fingerprint of fifteen batches of samples is shown in Figure 3. A total of 49 peaks were identified as common peaks; detailed information on them is given in Appendix A. The similarity of the fifteen batches of samples was calculated based on the “vector cosine” method, that was built in the same software using all the detected peaks [29]. The values are shown in Table 3. It shows that most of samples had a similarity level of 0.95–0.99, compared with the reference fingerprint.

However, when a clustering analysis was carried out, some instabilities were found. Based on the peak areas of the seven identified compounds (Appendix A), quality variations of the 15 batches of samples were evaluated by a clustering analysis using HemI 1.0 software [30]. The clustering analysis (Figure 4) shows that samples S1 to S10 have obvious distinctions with samples S11 to S15, although the similarity values of these samples are much closer. When checking each compound, we observed that the peak areas of liquiritin and glycyrrhizic acid in samples S11 to S15 were larger than those in samples S1 to S10. Considering that the two compounds both come from *Glycyrrhizae radix*, we can assume that some quality variations from this herb were introduced into the Siji-kangbingdu mixture in the manufacturing processes, so we recommended that the producers pay more attention to their *Glycyrrhizae radix*.

### 2.5. Quantitative Analysis of Siji-kangbingdu mixture by QAMS Method

The last step of our evaluation strategy was to determine the target compounds and provide quantitative information for the Siji-kangbingdu mixture. According to the UPLC-TripleTOF-MS experiment, seven abundant compounds which are of clinic significance were selected for content determination. The QAMS method was employed, because it’s simple and economical for quality control.

#### 2.5.1. Calculation of Relative Correction Factors (RCFs, ƒx)

In each UPLC chromatogram, RCFs were calculated using equation (2), as previously reported [31,32]:ƒx = (Cx × A_Forsythoside A_)/(C_Forsythoside A_ × Ax),(2)

In this equation, Cx indicates the concentration of a specific compound, and Ax indicate the peak areas of that compound. The most important step in calculating RCFs is to select the appropriate internal standard reference. In this work, forsythoside A was selected as the internal standard reference, considering its moderate retention time, stability, and obtainability. The RCFs of the other six compounds are shown in Table 4.

#### 2.5.2. Reproducibility Investigation of RCFs

The RCFs were calculated using different instruments (LC-20ADXR and Waters ACQUITY UPLC system), and with different columns, including XBridgeTMBEH column (Waters, Milford, MA, USA), ACQUITY UPLC^®^ BEH C18 column (Waters), and Accucore C18 column (Thermo scientific, Waltham, MA, USA). The values and variants are shown in Table 5; no significant differences were observed between the instruments and columns.

Additionally, we also tested the influence of different flow rates (0.38, 0.39, 0.41, 0.42, 0.43, and 0.45 mL/min), different injection volumes (0.2, 0.5, 1.0, 2.0, 3.0, 4.0, 5.0, and 6.0 μL) and different column temperatures (28, 29, 30, 31, 32, 33, and 35 °C) on RCFs. The results are shown in Table 6, Table 7 and Table 8. No significant difference was found between different flow rates, injection volumes, and column temperatures.

#### 2.5.3. Calculation of Relative Retention Time

The retention times or relative retention times (t) are important for peak location. In this study, the relative retention time was adopted for its stability and adaptability. Forsythoside A was selected as the internal standard reference, and data were calculated using different instruments and columns (Table 9). The results show that limited variance exists under different conditions. All these data indicate that the QAMS method established in this study is suitable for the quantitative analysis of Siji-kangbingdu mixture.

#### 2.5.4. Comparative Assessment between QAMS Method and External Standard Method (ESM)

In order to verify the reliability of the QAMS method, the concentrations of the seven compounds were calculated using the QAMS and ESM methods, respectively. The relative error (RE) was built to calculate the deviations between the two methods, as shown in equation (3):RE (%) = (Contents determined by QAMS method − Contents determined by ESM method) × 100%/Contents determined by ESM method),(3)

The results are shown in Table 10. The RE values are all less than 5%, which means that there was no significant difference in compound contents between the two methods.

## 3. Discussion

In TCM clinics, a large number of prescriptions are issued by from doctors every day; a number of these prescriptions are for hospital preparations or TCM patent medicines, due to their relatively reliable therapeutic effects. Like Siji-kangbingdu mixture, these preparations or patent medicines are numerous, and most of them lack qualitative and quantitative studies. So, finding a method by which to quickly and comprehensively evaluate their quality is a big problem. In this study, we report an evaluation strategy for fast, efficient, and economical quality evaluations of Siji-kangbingdu mixture, including overall chemical identification, fingerprint characterization, and multi-compound determination. UPLC-TripleTOF-MS technology, UPLC fingerprint technology, and the QAMS method were employed. This strategy provides large amounts of information on different levels, and can help us estimate the quality and stability of different batches of Siji-kangbingdu mixture within a single day.

UPLC gradient optimization is the most time-consuming part of this work. First, a large and fast gradient (0–10 min, 5–95% solvent B) were given to test the polarity distribution of all compounds; we found that most compounds were of a high polarity. Second, a series of slower and targeted gradients were designed to separate most of compounds. Third, some gradient intervals were shortened or prolonged to adapt the distribution intensity of the peaks, assigning more time for more similar compounds. Lastly, the gradient was validated with different mobile phases, flow rates, column temperatures, and sample loading volumes. After that, a complete UPLC gradient was generated.

For LC-MS analysis, the high resolution of mass detectors can effectively counteract the deficiency of LC systems; however, in this study, we still sought the best separation efficiency of the various compounds for two reasons: On the one hand, the elution gradient in this study was designed to be an all-purpose one, i.e., for chemical profiling, fingerprint analysis, and quantitative analysis. This strategy makes it possible for comparisons to be undertaken between chromatograms under different detectors, which is useful for compound identification. On the other hand, the higher resolution of the mass detector, the larger the number and the higher the quality of secondary mass spectra that can be obtained.

In this study, we did preexperiments of the *m/z* distribution of samples; no compounds larger than 1000 Da were identified. In *Platycodonis radix*, some platycodins existed with molecule weights of about 1200, but they were barely detected due to their low responses caused by their ionization properties. So, a mass detection range of 100–1000 *m/z* was adopted [33]. 

In the fingerprint experiments, we conducted a clustering analysis using 7 identified peaks to show the distinctions and similarities among samples. Additionally, we also performed a principal component analysis (PCA) using all 49 peaks. The results (Appendix A) show similar relationships among the samples but provide more detailed information. From the scatter plot, we can see that samples S1–10 differed considerably from samples S11–15, while distinctions are also obvious within the two sample clusters, especially for sample S3.

In this study, some details should be explained: Firstly, the preparation methods of negative sample solutions were based on those described in the literature and the single herb UPLC fingerprint. Simply, we tested all 11 herbs using the fingerprint method to check the existence of each compound in these herbs, after which the literature was referred to. As shown in the end of Section 4.1, we used a certain batch of each herbal slices for negative sample preparation for two reasons: first, Siji-kangbingdu mixture is made only by one company, i.e., the Shaanxi Haitian Pharmaceutical Co., Ltd. To our knowledge, this company has a fixed supplier of herbal slices, which implies fixed production areas for each of the herbal slices. The herbal slices used in this study were provided by the same supplier. Second, according to the literature, herbal slices from the same production area have relatively stable chemical constituents, differing only in the content ratios of compounds. So, we think that it is much reliable to use one batch of each of the herbal slices. Also, the selection of the internal standard reference in QAMS was according to the quality standards of the China Food and Drug Administration (CFDA), in which the forsythin in the herb *Forsythia Fructus* is selected as a marker compound of Siji-kangbingdu mixture for quality control. Because *Forsythia Fructus* is the “monarch drug” in Siji-kangbingdu mixture, we chose another phenylpropanoid compound, i.e., forsythoside A, as a reference material. This compound is cheap and stable, and also yields a good peak ratio with the other six compounds. Thirdly, for convenient practice, Siji-kangbingdu mixture was directly diluted in a methanol solution without any extraction process; this sample preparation method is economical and environmental protective.

## 4. Materials and Methods

### 4.1. Chemicals and Materials

Fifteen batches (S1–S15) of Siji-kangbingdu mixture were purchased from Shaanxi Haitian Pharmaceutical Co., Ltd. (Shaanxi, China). The detailed information of all samples is listed in Table 11. The standard materials, including chlorogenic acid, liquiritin, rutin, forsythoside A, and forsythin, were purchased from the National Institutes for Food and Drug Control (Beijing, China); Isochlorogenic acid A and glycyrrhizic acid were purchased from Chengdu Wei-ke-qi Biological Technology Co., Ltd. (Chengdu, Sichuan Province, China); information on them is provided in Table 12, and their chemical structures are shown in Figure 5. UPLC grade phosphoric acid was purchased from Tianjin Kermel Chemical Reagent Co., Ltd. (Tianjin), and acetonitrile and methanol were acquired from Honeywell Trading Co., Ltd. (Shanghai). The same methanol of UPLC grade was used in the sample preparations for the UPLC and LC-MS analysis, and the needle wash solutions were also prepared with this methanol (20% and 80% concentrations for weak and strong washes, respectively). The herbal slices used for negative sample preparation were purchased from the Shaanxi Xing-sheng-de Pharmaceutical Co., Ltd. (Tongchuan, Shaanxi Province, China); and the information of these herbal slices is provided in Table 13.

### 4.2. Instruments and Conditions

The chemical identification analysis was performed on an UPLC-MS system equipped with Waters ACQUITY UPLC system (Waters, Milford, MA, USA) tandem TripleTOF 5600+ mass spectrometer (AB Sciex, Framingham, MA, USA). Samples were first separated on an Accucore C18 column (2.6 μm, 100 × 2.1 mm), the mobile phase were water with 0.1% formic acid (A) and acetonitrile (B). The applied gradient was 0–7.5 min, 4% B; 7.5–8 min, 4–9.7% B; 8–20 min, 9.7–10% B; 20–21 min, 10–12% B; 21–21.5 min, 12–12.7% B; 21.5–30 min, 12.7–13% B; 30–35 min, 13–15.7% B; 35–40 min, 15.7–16% B; 40–40.5 min, 16–30% B; 40.5–50 min, 30% B; 50–51.5 min, 30–95% B; 51.5–55 min, 95% B; 55–57 min, 95–4% B; and 57–58 min, 4% B. The flow rate was 0.4 mL/min. The column temperature was set at 30 °C. The injection volume was 5 μL. The eluted fractions were then ionized using a DuoSpray ion source and detected using a mass spectrometer in positive and negative modes, respectively. The parameters were set as follows: source temperature: 500 °C, ion spray voltage floating: ±4500 V (for positive and negative ion mode respectively), curtain gas flow: 35 L/h, nebulizing gas (GS1): 50 L/h, drying gas (GS2): 50 L/h. Dynamic background subtract mode was on and the most intensive 10 precursor ions were selected for MS/MS detection. The de-clustering potential of Q1 and Q2 MS were both set at ±80 V. The collision energy in Q1 and Q2 MS were ±10 eV or ±35 eV, respectively. The TOF system was calibrated by the calibration module of the TripleTOF system, with calibration solutions (product code: PN4460131 and PN4460134) being provided along with the mass spectrometry. In this study, calibrations in positive and negative modes were conducted immediately before analysis.

In the fingerprint analysis and QAMS experiment, samples were separated on the LC-20ADXR (SHIMADZU, Japan) UPLC system with the system controller (CBM-20A), solution delivery unit (LC-20AD), a degasser (DGU-20A5R), a column thermostat (CTO-20AC), and a photodiode array UV-visible detector (SPD-M20A 230V). Samples were separated on an Accucore C18 column (2.6 μm, 100 × 2.1 mm). The mobile phase was water with 0.5% phosphoric acid (A) and acetonitrile (B). The elution gradient, flow rate, and column temperature were all the same as those in the UPLC-MS experiment. The injection volume was 3 μL, and the detection wavelength was 254 nm for all peaks.

### 4.3. Preparation of the Standard Solutions

The chlorogenic acid, liquiritin, rutin, forsythoside A, isochlorogenic acid A, forsythin, and glycyrrhizic acid were accurately weighed and dissolved in acetonitrile, making a mixture of 0.51 mg/mL of chlorogenic acid, 1.2 mg/mL liquiritin, 1.25 mg/mL rutin, 9.13 mg/mL forsythoside A, 0.25 mg/mL isochlorogenic acid A, 4.12 mg/mL forsythin, and 2.46 mg/mL glycyrrhizic acid. The mixed reference solution was diluted 3.5 to 112 times with 20% methanol to obtain six different concentrations of standard solutions. All standard solutions were stored in a refrigerator at 4 °C until use.

### 4.4. Preparation of Siji-Kangbingdu Mixture Sample Solutions and Negative Sample Soluitons

In the UPLC-TripleTOF-MS experiment, samples were diluted 10 times with 20% methanol, followed by filtering with a 0.22 μm membrane filter and loading onto the column. Otherwise, samples were diluted 2 times with 20% methanol and filtered with a 0.22 μm membrane before use. Negative control samples were produced according to the quality standard of Siji-Kangbingdu Mixture prescribed by CFDA, with or without specific herbs. Simply, *Houttuyniae Herba* (120 g), *Platycodonis Radix* (120 g), *Mori Folium* (120 g), *Forsythiae Fructus* (120 g), *Schizonepetae Herba* (50 g), *Menthae Haplocalycis Herba* (60 g), *Perillae Folium* (60 g), *Armeniacae Semen Amarum* (50 g), *Phragmitis Rhizoma* (120 g), *Chrysanthemi Flos* (100 g), and *Glycyrrhizae Radix et Rhizoma* (50 g) were extracted with boiling water (volatile oils were also gathered from the vapor and combined with the extraction solution), and the extraction solution was treated with 60% ethanol overnight for the precipitation of macromolecules; the supernatant was isolated, sterilized, and diluted to 1000 mL.

### 4.5. Statistical Analysis

The LC-MS data were handled using the Analyst TF 1.7 software (AB Sciex, Framingham, MA, USA) and embedded with PeakView Extra Utilities (2.2.0.11391, AB Sciex, Framingham, USA), and MasterView (1.1.1944.0, AB Sciex, Framingham, USA). In the compound identification analysis, MS and MS/MS data were both used in the “candidate search” algorithm, used to search against the TCM Library 1.0 C/D/V and TCM MS/MS library [34,35]. The following parameters were set to ensure the accuracy of each identified compound: mass error < 5 ppm, isotope ratio difference < 10%, and library score > 30. 

The UPLC fingerprint analysis was performed using the “Similarity Evaluation System of Traditional Chinese Medicine Chromatographic Fingerprint” software (2012.130723 edition, Chinese Pharmacopoeia Commission, Beijing, China), peaks were manually marked, automatic corrected, and aligned, followed by reference/sample fingerprint generation. The heatmap and clustering analysis were performed using the HemI 1.0 software (Huazhong University of Science and Technology, Wuhan, Hubei Province, China). The PCA analysis was performed by using SIMCA-P 11.0 software (Umetrics, Umeå, Västerbotten, Sweden). For quantitative analysis, the peak areas were integrated using the LC solutions software (SHIMADZU, Tokyo, Japan), and linearity was calculated using the EXCEL software.

## 5. Conclusions

In this study, an evaluation strategy was integrated with UPLC-TripleTOF-MS identification, UPLC fingerprint analysis, and quantitation method of QAMS. This method comprehensively uncovered the qualities of Siji-kangbingdu mixture. Seventy compounds were identified as the material basis of Siji-kangbingdu mixture, which will facilitate pharmacokinetic and pharmacological research carried out on this mixture. The UPLC fingerprint and clustering analysis revealed both the quality stability and variations, reminding manufactures to standardize the quality of herbal slices in the feeding process. The determination of seven active compounds further characterized the quality of the mixture. In conclusion, the established method is comprehensive, reliable, and economical. It provides a reference for quality evaluations of Siji-kangbingdu mixture and other TCM patent medicines.

## Figures and Tables

**Figure 1 molecules-24-03545-f001:**
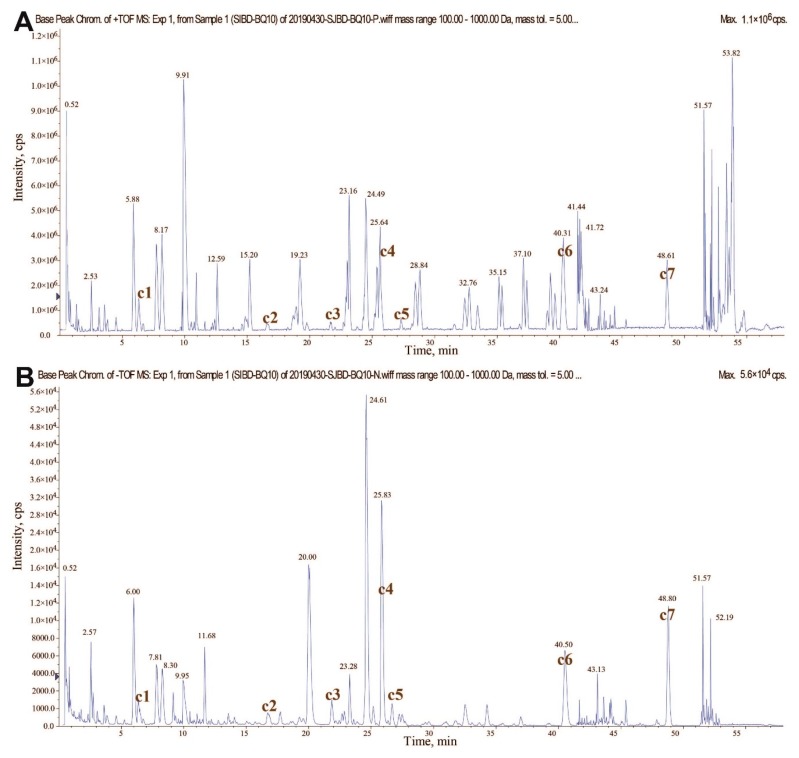
Base peak ion chromatograms of Siji-kangbingdu mixture detected using UPLC-TripleTOF-MS technology. (**A**) Detected in positive ion mode; (**B**) Detected in negative ion mode. In the two panels, the label c1 to c7 represent different compounds: c1—chlorogenic acid, c2—liquiritin, c3—rutin, c4—forsythoside A, c5—isochlorogenic acid A, c6—forsythin, and c7—glycyrrhizic acid.

**Figure 2 molecules-24-03545-f002:**
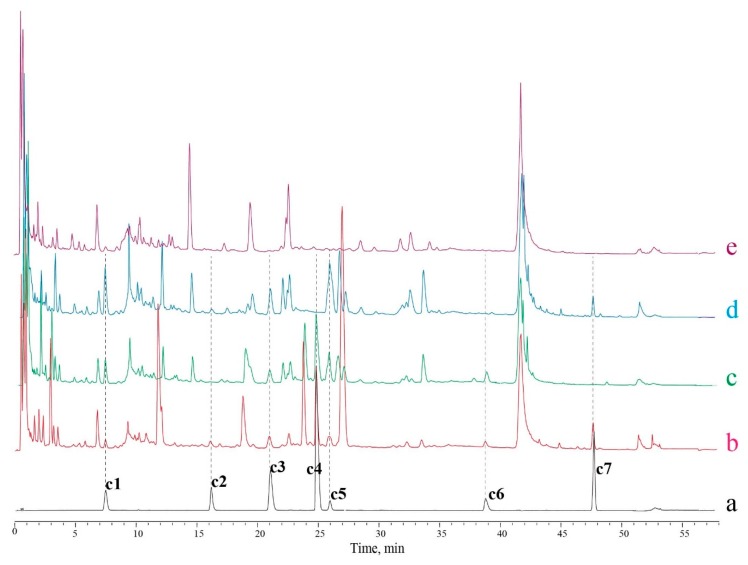
The UPLC chromatogram of standard solutions, samples and negative control samples. (**a**): Mixed standards solution; (**b**): Sample S1; (**c**): Negative control sample of *Glycyrrhizae Radix* deficiency; (**d**): Negative control sample of *Forsythiae Fructus* deficiency; (**e**): Negative control sample of *Mori Folium*, *Chrysanthemi Flos*, *Houttuyniae Herba*, *Forsythiae Fructus*, and *Glycyrrhizae Radix et Rhizoma* multi-deficiency. c1—chlorogenic acid, c2—liquiritin, c3—rutin, c4—forsythoside A, c5—isochlorogenic acid A, c6—forsythin, and c7—glycyrrhizic acid.

**Figure 3 molecules-24-03545-f003:**
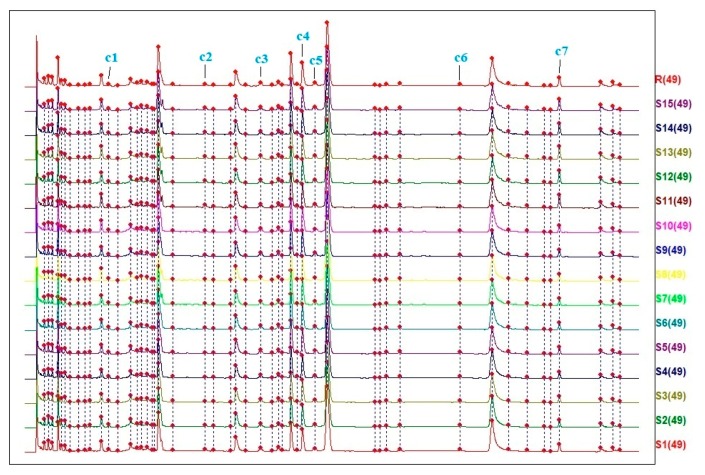
The UPLC fingerprint of fifteen batches of samples S1–S15. A common pattern was extracted and used as reference fingerprint (R). Among these 49 common peaks: c1—chlorogenic acid (peak 12); c2—liquiritin (peak 22); c3—rutin (peak 27); c4—forsythoside A (peak 33); c5—isochlorogenic acid A (peak 34); c6—forsythin (peak 40); c7—glycyrrhizic acid (peak 46).

**Figure 4 molecules-24-03545-f004:**
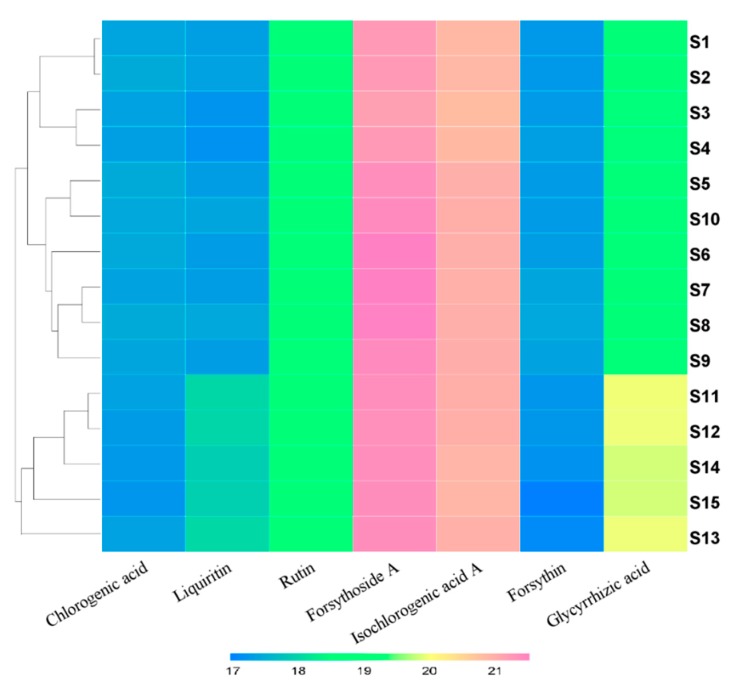
Heatmap of peak areas from 15 batches of samples. The HemI software was used to generate this map, and the hierarchical method was used for clustering against different samples. Peak area values were normalized by log2 transformation, and represented with different colors in the color gradation at the bottom.

**Figure 5 molecules-24-03545-f005:**
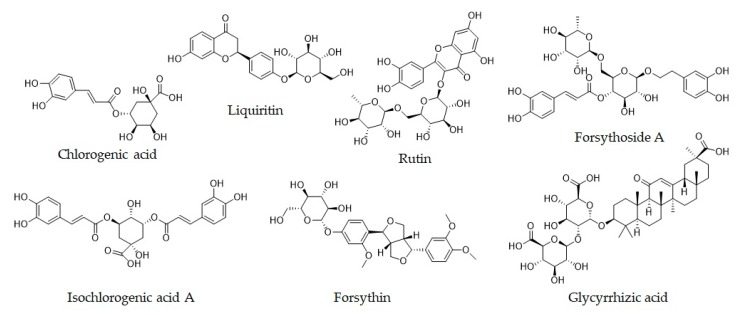
Chemical structures of seven determined compounds.

**Table 1 molecules-24-03545-t001:** The regression equations, Limit of Detections (LODs) and Limit of Quantity (LOQs) of seven compounds.

Compounds	Regression Equations	Linear Range (μg/mL)	R^2^	LODs (μg/mL)	LOQs (μg/mL)
Chlorogenic acid	Y = 2 × 10^7^X + 2015.5	4.554~145.7	1.0000	0.0861	0.2869
Liquiritin	Y = 6 × 10^6^X − 2903.9	10.71~342.8	1.0000	0.0701	0.2335
Rutin	Y = 2 × 10^7^X − 1765.6	11.16~357.1	1.0000	0.1003	0.3342
Forsythoside A	Y = 8 × 10^6^X + 1164.7	81.52~2608	1.0000	0.0464	0.1546
Isochlorogenic acid A	Y = 2 × 10^7^X − 1940.5	2.232~71.43	1.0000	0.1034	0.3720
Forsythin	Y = 1 × 10^6^X + 11002	36.79~1177	0.9998	0.0287	0.0956
Glycyrrhizic acid	Y = 9 × 10^6^X + 176.68	21.96~702.9	1.0000	0.8994	3.6607

**Table 2 molecules-24-03545-t002:** The precision, stability, repeatability, and accuracy of the analytical method for seven compounds.

Compounds	Precision RSD (%)	Stability RSD (%)	Repeatability RSD (%)	Accuracy	
				Mean (%)	RSD (%)
Chlorogenic acid	2.80	2.63	1.52	99.60	2.11
Liquiritin	3.20	2.64	3.94	95.11	0.34
Rutin	1.20	1.06	0.85	104.19	1.83
Forsythoside A	0.42	0.32	0.61	90.87	2.66
Isochlorogenic acid A	3.20	1.77	3.79	99.47	1.70
Forsythin	0.64	0.98	3.45	100.76	1.86
Glycyrrhizic acid	0.14	0.14	0.36	95.00	4.24

**Table 3 molecules-24-03545-t003:** The Similarity of fifteen batches of samples.

No.	Similarity	Percentage of Non-Common Peak Area
S1	0.984	16.23
S2	0.984	14.88
S3	0.957	14.08
S4	0.956	14.46
S5	0.950	14.12
S6	0.935	8.46
S7	0.976	15.71
S8	0.982	15.80
S9	0.977	14.96
S10	0.982	14.19
S11	0.980	16.79
S12	0.965	16.98
S13	0.980	16.46
S14	0.969	17.53
S15	0.982	16.24

Note: The similarity value was calculated based on the “vector cosine” method.

**Table 4 molecules-24-03545-t004:** Relative correction factors (RCFs) of six compounds against forsythoside A.

Instrument	Column	RCF Values	
LC-20ADXR	Accucore C18 column (2.6 μm,100 × 2.1 mm, Thermo scientific)	ƒ_c1/c4_	0.4860
		ƒ_c2/c4_	1.1992
		ƒ_c3/c4_	0.4872
		ƒ_c5/c4_	0.4852
		ƒ_c6/c4_	5.0971
		ƒ_c7/c4_	0.8563

Note: In this table, c1—chlorogenic acid, c2—liquiritin, c3—rutin, c4—forsythoside A, c5—isochlorogenic acid A, c6—forsythin, and c7—glycyrrhizic acid.

**Table 5 molecules-24-03545-t005:** RCFs in difference instruments and columns.

Instrument	Column	ƒ_c1/c4_	ƒ_c2/c4_	ƒ_c3/c4_	ƒ_c5/c4_	ƒ_c6/c4_	ƒ_c7/c4_
LC-20ADXR	XBridgeTMBEH (150 × 2.1 mm, 2.5 μm)	0.4761	1.1711	0.4869	0.4817	5.7452	1.0605
	ACQUITY UPLC^®^ BEH C18 (100 × 2.1 mm, 1.7 μm)	0.4768	1.2412	0.4695	0.4822	5.3664	1.0167
	Accucore C18 (100 × 2.1 mm, 2.6 μm)	0.4729	1.1699	0.4836	0.4877	5.9337	1.0145
Waters ACQUITY UPLC system	XBridgeTMBEH (150 × 2.1 mm, 2.5 μm)	0.4962	1.2409	0.4660	0.4813	5.7142	1.0712
	ACQUITY UPLC^®^ BEH C18 (100 × 2.1 mm, 1.7 μm)	0.4896	1.1736	0.4854	0.4813	5.4095	0.9405
	Accucore C18 (100 × 2.1 mm, 2.6 μm)	0.4929	1.2445	0.4726	0.4679	5.6470	0.9422
	Means	0.4841	1.2069	0.4773	0.4804	5.6360	1.0076
	RSD (%)	2.06	3.21	1.89	1.37	3.81	5.57

Note: In this table, c1—chlorogenic acid, c2—liquiritin, c3—rutin, c4—forsythoside A, c5—isochlorogenic acid A, c6—forsythin, and c7—glycyrrhizic acid.

**Table 6 molecules-24-03545-t006:** RCFs in difference flow rates.

Flow Rates (mL/min)	ƒ_c1/c4_	ƒ_c2/c4_	ƒ_c3/c4_	ƒ_c5/c4_	ƒ_c6/c4_	ƒ_c7/c4_
0.38	0.4899	1.1855	0.4853	0.4859	5.0013	0.8557
0.39	0.4850	1.1820	0.4857	0.4859	5.0234	0.8547
0.41	0.4892	1.1683	0.4855	0.4878	5.0544	0.8537
0.42	0.4913	1.1866	0.4861	0.4896	5.0579	0.8537
0.43	0.4943	1.1734	0.4869	0.4789	5.0654	0.8534
0.45	0.4914	1.1796	0.4836	0.4787	5.0895	0.8521
Means	0.4902	1.1792	0.4855	0.4844	5.0487	0.8539
RSD (%)	0.62	0.61	0.23	0.95	0.62	0.14

Note: In this table, c1—chlorogenic acid, c2—liquiritin, c3—rutin, c4—forsythoside A, c5—isochlorogenic acid A, c6—forsythin, and c7—glycyrrhizic acid.

**Table 7 molecules-24-03545-t007:** RCFs in difference injection volumes.

Injection Volume (μL)	ƒ_c1/c4_	ƒ_c2/c4_	ƒ_c3/c4_	ƒ_c5/c4_	ƒ_c6/c4_	ƒ_c7/c4_
0.2	0.4928	1.1718	0.4936	0.4838	4.8557	0.8565
0.5	0.4925	1.1895	0.4887	0.5191	4.9608	0.8528
1	0.4682	1.1847	0.4842	0.4822	5.0616	0.8516
2	0.4883	1.1722	0.4852	0.4878	5.2151	0.8515
3	0.4863	1.1699	0.4843	0.4857	5.3242	0.8518
4	0.4902	1.1704	0.4841	0.4804	5.3746	0.8530
5	0.4851	1.1658	0.4843	0.4822	5.4040	0.8533
6	0.4889	1.1710	0.4851	0.4792	5.4585	0.8541
Means	0.4866	1.1744	0.4862	0.4876	5.2068	0.8531
RSD (%)	1.62	0.70	0.69	2.68	4.29	0.19

Note: In this table, c1—chlorogenic acid, c2—liquiritin, c3—rutin, c4—forsythoside A, c5—isochlorogenic acid A, c6—forsythin, and c7—glycyrrhizic acid.

**Table 8 molecules-24-03545-t008:** RCFs in difference column temperatures.

Column Temperature (°C)	ƒ_c1/c4_	ƒ_c2/c4_	ƒ_c3/c4_	ƒ_c5/c4_	ƒ_c6/c4_	ƒ_c7/c4_
28	0.4912	1.1719	0.4849	0.4915	5.0145	0.8536
29	0.4854	1.1922	0.4857	0.4872	5.0184	0.8549
30	0.4951	1.1719	0.4851	0.4937	5.0272	0.8532
31	0.4880	1.1894	0.4898	0.4817	5.0303	0.8546
32	0.4880	1.1894	0.4898	0.4817	5.0303	0.8546
33	0.4912	1.1838	0.4886	0.4832	5.0215	0.8537
35	0.4913	1.1821	0.4858	0.4914	5.0267	0.8533
Means	0.4900	1.1830	0.4871	0.4872	5.0241	0.8540
RSD (%)	0.64	0.70	0.46	1.04	0.12	0.08

Note: In this table, c1—chlorogenic acid, c2—liquiritin, c3—rutin, c4—forsythoside A, c5—isochlorogenic acid A, c6—forsythin, and c7—glycyrrhizic acid.

**Table 9 molecules-24-03545-t009:** The relative retention time of seven compounds.

Instrument	Column	t_c1/c4_	t_c2/c4_	t_c3/c4_	t_c5/c4_	t_c6/c4_	t_c7/c4_
LC-20ADXR	XBridgeTMBEH (150 × 2.1 mm, 2.5 μm)	0.3266	0.7457	0.8031	1.0795	1.4729	1.7978
	ACQUITY UPLC^®^ BEH C18 (100 × 2.1 mm, 1.7 μm)	0.3120	0.7207	0.7781	1.0767	1.4735	1.8311
	Accucore C18 (100 × 2.1 mm, 2.6 μm)	0.3355	0.7226	0.8161	1.0770	1.4618	1.8072
Waters Corporation	XBridgeTMBEH (150 × 2.1 mm, 2.5 μm)	0.3170	0.6850	0.8006	1.0249	1.4906	1.8084
	ACQUITY UPLC^®^ BEH C18 (100 × 2.1 mm, 1.7 μm)	0.3073	0.6584	0.8559	1.0756	1.4719	1.8065
	Accucore C18 (100 × 2.1 mm, 2.6 μm)	0.2969	0.6528	0.8671	1.0801	1.5891	1.7856
	Means	0.3158	0.6975	0.8201	1.0689	1.4933	1.8061
	RSD (%)	4.36	5.42	4.21	2.03	3.21	0.83

Note: In this table, c1—chlorogenic acid, c2—liquiritin, c3—rutin, c4—forsythoside A, c5—isochlorogenic acid A, c6—forsythin, and c7—glycyrrhizic acid.

**Table 10 molecules-24-03545-t010:** The contents of seven compounds in fifteen batches of samples, as determined with the ESM and QAMS methods (μg/mL).

No.	Forsythoside A	Chlorogenic Acid			Liquiritin			Rutin			Isochlorogenic Acid A			Forsythin			Glycyrrhizic Acid		
	ESM	ESM	QAMS	RE%	ESM	QAMS	RE%	ESM	QAMS	RE%	ESM	QAMS	RE%	ESM	QAMS	RE%	ESM	QAMS	RE%
S1	540.9	24.3	23.4	−3.69	56.6	55.5	−1.99	57.3	55.3	−3.49	259.9	262.4	0.95	235.4	228.1	−3.10	132.9	128.0	−3.69
S2	545.4	25.3	24.4	−3.73	57.8	56.6	−1.95	57.9	55.9	−3.48	262.1	264.6	0.95	234.9	227.6	−3.08	132.5	127.6	−3.69
S3	527.7	23.6	22.7	−3.66	52.3	51.2	−2.11	57.7	55.7	−3.49	253.6	256.0	0.95	237.2	229.7	−3.14	124.6	120.0	−3.69
S4	534.3	22.9	22.1	−3.63	51.0	49.9	−2.15	57.3	55.3	−3.49	256.7	259.2	0.95	246.0	237.7	−3.35	124.6	120.0	−3.69
S5	589.0	25.5	24.5	−3.73	56.5	55.3	−1.99	68.5	66.2	−3.42	283.0	285.7	0.96	238.4	230.9	−3.17	133.5	128.5	−3.69
S6	587.7	25.1	24.1	−3.71	55.7	54.5	−2.01	68.7	66.4	−3.42	282.4	285.1	0.96	243.9	235.9	−3.30	133.9	129.0	−3.69
S7	586.8	23.8	23.0	−3.67	56.4	55.3	−1.99	66.3	64.0	−3.43	281.9	284.7	0.96	259.6	250.1	−3.65	133.1	128.2	−3.69
S8	590.2	25.3	24.4	−3.72	61.0	59.8	−1.87	67.7	65.4	−3.43	283.6	286.3	0.96	263.7	253.8	−3.73	132.9	128.0	−3.69
S9	600.8	24.4	23.5	−3.69	56.5	55.4	−1.98	66.9	64.6	−3.43	288.7	291.5	0.96	254.6	245.6	−3.54	134.3	129.3	−3.69
S10	592.9	24.9	23.9	−3.71	59.9	58.7	−1.90	68.1	65.8	−3.42	284.9	287.7	0.96	239.1	231.5	−3.18	134.3	129.3	−3.69
S11	594.4	23.6	22.8	−3.66	90.3	89.0	−1.41	62.9	60.7	−3.45	285.6	288.4	0.96	229.0	222.3	−2.92	254.9	245.5	−3.70
S12	594.0	22.4	21.6	−3.60	89.7	88.4	−1.42	63.1	60.9	−3.45	285.4	288.2	0.96	230.1	223.3	−2.95	254.1	244.7	−3.70
S13	582.4	23.5	22.6	−3.65	91.2	89.9	−1.40	76.0	73.4	−3.39	279.8	282.5	0.96	208.4	203.6	−2.32	253.2	243.8	−3.70
S14	555.1	21.8	21.0	−3.58	84.2	82.9	−1.48	60.0	57.9	−3.47	266.8	269.3	0.96	220.4	214.4	−2.69	232.6	224.0	−3.70
S15	550.4	21.4	20.6	−3.55	84.7	83.4	−1.48	60.0	58.0	−3.47	264.5	267.1	0.96	189.1	186.0	−1.64	232.5	223.9	−3.70

**Table 11 molecules-24-03545-t011:** Information on the fifteen batches of samples.

NO.	Voucher Number	NO.	Voucher Number
S1	181060	S9	190302
S2	181119	S10	190543
S3	181115	S11	190172
S4	181118	S12	190173
S5	180357	S13	190174
S6	190120	S14	190176
S7	190121	S15	190177
S8	190123		

**Table 12 molecules-24-03545-t012:** Information on the standard materials.

Standard Materials	Purity	Lot Number
Chlorogenic acid	≥96.8%	110753-201817
Liquiritin	≥93.1%	111610-201607
Rutin	≥91.7%	100080-201811
Forsythoside A	≥97.2%	111810-201707
Isochlorogenic acid A	≥98%	wkq18041207
Forsythin	≥95.1%	110821-201816
Glycyrrhizic acid	≥98%	wkq18011803

**Table 13 molecules-24-03545-t013:** Information on the of herbal slices used in the negative sample preparation.

Herbal Slice Name	Original Plant Name	Collection Location	Voucher/Lot Number
*Houttuyniae Herba*	*Houttuynia cordata* Thunb.	Sichuan	20181201
*Platycodonis Radix*	*Platycodon grandiflorum* (Jacq.) A.DC.	Shaanxi	20190201
*Mori Folium*	*Morus alba* L.	Jiangsu	20181201
*Forsythiae Fructus*	*Forsythia suspensa* (Thunb.) Vahl	Shanxi	20180801
*Schizonepetae Herba*	*Schizonepeta tenuifolia* Briq.	Shaanxi	20190301
*Menthae Haplocalycis Herba*	*Mentha haplocalyx* Briq.	Hebei	20190401
*Perillae Folium*	*Perilla frutescens* (L.) Britt.	Guangxi	20190401
*Armeniacae Semen Amarum*	*Prunus armeniaca L.var.ansu* Maxim.	Shaanxi	20190501
*Phragmitis Rhizoma*	*Phragmites communis* Trin.	Shaanxi	20190501
*Chrysanthemi Flos*	*Chrysanthemum morifolium* Ramat.	Zhejiang	20190301
*Glycyrrhizae Radix et Rhizoma*	*Glycyrrhiza uralensis* Fisch.	Gansu	20190302

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
