# Peer review of "Multi-Evaluating Strategy for Siji-kangbingdu Mixture: Chemical Profiling, Fingerprint Characterization, and Quantitative Analysis"

_molecules, 2019, doi:10.3390/molecules24193545_

Round 1

Reviewer 1 Report

The work is valuable, well written and is suitable for publication in Molecules, but some doubts must be clarified:
1. "Then chromatograms were subjected to the" Similarity Evaluation System of 188 Traditional Chinese Medicine Chromatographic Fingerprint "" - what does the statement chromatograms mean? In what form were the chromatograms processed to generate a fingerprint?
2. How was the gradient optimized in UPLC?
3. What do the values in the 2nd column of Table 3 mean, are they correlation coefficients or any other?
4. In my opinion, in addition to cluster analysis, the authors should perform the principal components analysis and attach the results of this analysis to the discussion (even in additional materials).
PCA analysis is widely used in fingerprint analysis and gives a large amount of information regarding sample similarity.
5. Fig 4 - no explanation as to what each color means on the heat map.

Author Response

Response to reviewer 1 comments

"Then chromatograms were subjected to the" Similarity Evaluation System of 188 Traditional Chinese Medicine Chromatographic Fingerprint "" - what does the statement chromatograms mean? In what form were the chromatograms processed to generate a fingerprint?

Response: The statement “chromatograms” actually means chromatogram data (CDF format) from the LC-20ADXR UPLC system, these data were imported into the fingerprint software which generated the fingerprint. We have made it clear in the manuscript (line 206-208).

How was the gradient optimized in UPLC?

Response: In section 2.1, we have already described the optimization of sample dilution solutions, mobile phase, column temperature, detection wavelength and etc. The gradient optimization is the most time-consuming part of this work, however, we have some experiences. Firstly, a large and fast gradient (0- 10 min, 5%-95% solvent B) were given to test the polarity distribution of all compounds, and we found most compounds were of high polarity. Second, a series of slower and targeted gradient were designed to separate most of compounds. Thirdly, some gradient intervals were shortened or prolonged to adapt the distribution intensity of peaks, assigning more time for high-similar compounds. Lastly, the gradient was validated with different mobile phases, flow rates, column temperatures and sample loading volumes. After that, a complete UPLC gradient was generated. We have added this detailed optimization steps in the Discussion section (line 309-316).

What do the values in the 2nd column of Table 3 mean, are they correlation coefficients or any other?

Response: In Table 3, the values in the 2nd column similarity values calculated by vector cosine algorithm, which is built in the “Similarity Evaluation System of Traditional Chinese Medicine Chromatographic Fingerprint” software. We have made it clear in the footnote of Table 3, citation is also given (line 223).

In my opinion, in addition to cluster analysis, the authors should perform the principal components analysis and attach the results of this analysis to the discussion (even in additional materials). PCA analysis is widely used in fingerprint analysis and gives a large amount of information regarding sample similarity

Response: The principal component analysis have been performed, results are added in the Discussion section (line 328-333) and supplementary materials (Figure S1).

Fig 4 - no explanation as to what each color means on the heat map.

Response: We have made it clear by the statements below Figure 4 in line 236-238.

Reviewer 2 Report

MAIN

1
The ratio of constituents listed in lines 37-39 remains hidden through all the work. -> Clear this in Intro or in Materials and Methods.

2
If the Siji-kangbingdu recipe is covered with 'patent' -> Cite the patent document appropriately, in Intro. Else, clear.

3
At the end, the reader don't even know if the Siji-kangbingdu mixture is a mixture of herbs or any preparation (e.g., in a liquid form). -> Clear this in Intro.

4
How is the Siji-kangbingdu mixture produced to meet the market? Is it a kind of on-prescription-drug? Is it produced / mingled immediately in pharmacy or could it be bought as a factory-derived preparation. -> Broaden the Intro and clear some things.

l.42a
What are the quality standards for common TCM formulas? -> Clear. Cite appropriate work / law doc.

l.42b
What are the legal aspects for higher quality standards in this particular case? -> Clear comprehensively. Cite appropriate work / law doc.

l.79-83
What is the reason to completely separate the compounds if HRMS is used as the postchromatographic detector? Are there numeorus isobaric compounds? Do you need to develop a method for simple UV detector? -> Clear to the readers.

l.110-111
How should be the reader assured that confidence level is enough for each compound?
'library' unprecised -> Clear + Cite
(at least at Materials and Methods, ch.4)

l.131
! there is an authors' note to cite sth that was not cited properly

l.121, Fig.1a-b
Why the range of detection was limited by 1000m/z? -> At least compounds from Platycodonis radix give larger ions. -> Clear.

5
The same 'leading' compounds are reported under different acronyms/abbreviations through all the work. See Fig.2 (as numbers) vs. Fig.3 (as other numbers) vs. Tab.5-9 (as letters) -> Homogenize this unless acceptable to publish.

6
Were the single-herb-fingerprints (l.287) based on one batch of each herbal substance (Tab.13)? Are such fingerprints reliable? -> Supply this in Materials and Methods or clear enough.

l.315, Fig.5.
-> Homogenize the stereochemistry in all compounds (e.g., some bonds not defined in isochlorogenic acid, forsythin; check bonds in glycyrrhizic acid)

ch.4.1
Concentrations/DER/DSR and solvents for UPLC samples are not defined. -> Define.

ch.4.2
How was the TOF calibrated? How frequently? -> Supply.

Table S1
How to explain formation of adducts with NH3 in acidic eluent?

LANGUAGE

The article needs language care. See some examples below.

l.18, 40
'anti-inflammation'

l.20
multiple meaning of 'its'

l.31
'mixtrue'

l.42

l.46, 54
'patant'

l.47, 103, 131, 289
'literatures'

l.84-85
What does mean 'diluted with methanol solutions (1:1 v/v)'? -> Clear. Remember that readers still don't know if subject is a mixture of dried/minced/sliced/fresh herbs or a liquid sample.

l.86
'solutions' ... 'was investigated'

l.119
'significances'

l.121 Fig.1 caption
'chromatography' -> 'chromatograms'?

l.134
"knocked out herbs" is pictorial but not clear enough

l.157
-> Check the superscripts in parameters of equations at the Table 1

l.197
'commom'

l.317, Tab.13
-> In plant names, remove italics from botanist acronyms (e.g., <i> Houttuynia cordata </i> Thunb. instead of <i> Houttuynia cordata Thunb. </i>).

l.334
'V' or 'eV'?

l.351
'before filtered'

Table S2
numerous 'aera'

Author Response

Response to reviewer 2 comments

The ratio of constituents listed in lines 37-39 remains hidden through all the work. -> Clear this in Intro or in Materials and Methods.

Response: The constituents have been detailed in section 4.4 (line 414-420).

If the Siji-kangbingdu recipe is covered with 'patent' -> Cite the patent document appropriately, in Intro. Else, clear.

Response: The Siji-kangbingdu recipe is covered by a Chinese patent, the citation have been added in the Introduction section (line 39).

At the end, the reader don't even know if the Siji-kangbingdu mixture is a mixture of herbs or any preparation (e.g., in a liquid form). -> Clear this in Intro.

Response: The Siji-kangbingdu Mixture is a Chinese herbal preparation in liquid form, containing 11 herbs. See line 39-40.

How is the Siji-kangbingdu mixture produced to meet the market? Is it a kind of on-prescription-drug? Is it produced / mingled immediately in pharmacy or could it be bought as a factory-derived preparation. -> Broaden the Intro and clear some things.

Response: We have made it clear enough in the Introduction section (line 42-45).

l.42a

What are the quality standards for common TCM formulas? -> Clear. Cite appropriate work / law doc.

l.42b

What are the legal aspects for higher quality standards in this particular case? -> Clear comprehensively. Cite appropriate work / law doc.

Response: We have added more information in the Introduction section and citations are provided (line 49-54).

l.79-83

What is the reason to completely separate the compounds if HRMS is used as the post-chromatographic detector? Are there numerous isobaric compounds? Do you need to develop a method for simple UV detector? -> Clear to the readers.

Response: We have explained some reasons for a complete UPLC separation method for LC-MS analysis in the Discussion section (line 317-323).

l.110-111

How should be the reader assured that confidence level is enough for each compound?

'library' unprecised -> Clear + Cite (at least at Materials and Methods, ch.4)

Response: We give some detailed explanations in section 2.2 (line 123-129), citations are given both in section 2.2 and section 4.5 (line 425).

l.131

! there is an authors' note to cite sth that was not cited properly

Response: We are sorry for this mistake! The citations have been correctly inserted in line 149.

l.121, Fig.1a-b

Why the range of detection was limited by 1000m/z? -> At least compounds from Platycodonis radix give larger ions. -> Clear.

Response: In this study, we did pre-experiments about m/z distribution of samples, no compounds larger than 1000 Da were identified. In Platycodonis radix, some platycodins exist with molecule weight of about 1200, but they were hardly detected, due to their low responses caused by ionization properties. So, the mass detection range of 100- 1000 m/z was adopted. Some explanations have been added in the Discussion section with literature cited (line 324-327).

The same 'leading' compounds are reported under different acronyms/abbreviations through all the work. See Fig.2 (as numbers) vs. Fig.3 (as other numbers) vs. Tab.5-9 (as letters) -> Homogenize this unless acceptable to publish.

Response: In Figure 1, Figure 2, Figure 3 and Table 5-9, the compounds have been represented with characters c1 to c7. Particularly in Figure 3, peak numbers were removed, while provided in the figure notes. See in line 138-142, line 159-165, line 217-221, and line 254-286.

Were the single-herb-fingerprints (l.287) based on one batch of each herbal substance (Tab.13)? Are such fingerprints reliable? -> Supply this in Materials and Methods or clear enough.

Response: As showed in Table 13, we used certain batch of each herbal slices for negative sample preparation for two reasons: One reason is that the Siji-kangbingdu Mixture is made only by one company, Shaanxi Haitian Pharmaceutical Co., Ltd. To our knowledge, this company has fixed supplier of herbal slices, which will lead to fixed producing area of each herbal slices, and the herbal slices used in this study were provided by the same supplier. Another reason is that, according to literatures, herbal slices of the same producing area have relatively stable chemical constituents, only differed in content ratio of compounds. So we think it is much reliable for us to use one batch of each herbal slices. Such explanations have been added in the Discussion section (line 337-344).

l.315, Fig.5.

-> Homogenize the stereochemistry in all compounds (e.g., some bonds not defined in

isochlorogenic acid, forsythin; check bonds in glycyrrhizic acid)

Response: Figure 5 have been re-constructed as showed in line 372.

ch.4.1

Concentrations/DER/DSR and solvents for UPLC samples are not defined. -> Define.

Response: The solvents and their concentrations used in sample preparation and needle wash are defined in section 4.1 (line 363-365).

ch.4.2

How was the TOF calibrated? How frequently? -> Supply.

Response: The TOF system were calibrated by the calibration module of the TripleTOF system, with calibration solutions provided along with the mass spectrometry. In this study, calibrations in positive and negative mode were conducted just before analysis. These statements have been added in section 4.2 (line 391-394).

Table S1

How to explain formation of adducts with NH3 in acidic eluent?

Response: Actually, in Table S1, contents in the “Formula” and “Identified compound” column were both inherited from the compound library. Take the compound of NO.15 as an example, the name “Salidroside +NH3” actually means the “Salidroside + [NH4]+” adducts; and the formula “C14H20O7.NH3” actually means “C14H20O7.NH4”. In TCM compounds, some glycosides have intensive tendencies to form the ammonium adducts rather than proton adducts. Although we did not add ammonium to the eluent, the ammonium ion may be brought in by the solvent system. We have made it clear in the table notes of Table S1.

LANGUAGE

The article needs language care. See some examples below.

l.18, 40 'anti-inflammation'  

l.20 multiple meaning of 'its'  

l.31 'mixtrue'  

l.42 l.46, 54 'patant'

l.47, 103, 131, 289 'literatures'  

l.84-85 What does mean 'diluted with methanol solutions (1:1 v/v)'? -> Clear. Remember that readers still don't know if subject is a mixture of dried/minced/sliced/fresh herbs or a liquid sample.   

l.86 'solutions' ... 'was investigated'   

l.119 'significances'   

l.121 Fig.1 caption 'chromatography' -> 'chromatograms'?   

l.134 "knocked out herbs" is pictorial but not clear enough   

l.157 -> Check the superscripts in parameters of equations at the Table 1   

l.197 'commom'   

l.317, Tab.13 ->In plant names, remove italics from botanist acronyms   

l.334 'V' or 'eV'?   

l.351 'before filtered'

Table S2 numerous 'aera'

Response: We have thoroughly checked this manuscript and corrected all grammatical mistakes, these revisions can be easily recognized by their color.